# The effect of biosynthesized silver nanoparticles on *Pseudomonas aeruginosa*-infected dogs wounds

**Ali Hussein Aldujaily**[1]*, **Kifah Fadhil Hassoon**[2], **Abdulameer Abid Hatem**[1],
**Murtadha Abbas**[3]

**1** Department of Veterinary Clinical Sciences, Faculty of Veterinary Medicine, University of Kufa, Kufa, Iraq,
**2** Department of Veterinary Microbiology, Faculty of Veterinary Medicine, University of Kufa, Kufa, Iraq,
**3** Department of Public Health, Faculty of Veterinary Medicine, University of Kufa, Kufa, Iraq

◔ All These authors contributed equally to this work.
* Alih.aldujaily@uokufa.edu.iq

## Abstract

Environmental factors such as lesion and bacterial infection risk affect wound healing. Many researchers are interested in silver nanoparticles (AgNPs) due to their anti-inflammatory, antibacterial, and wound-healing properties. This study compared AgNPs to Amikacin on dogs' *P. aeruginosa*-infected wounds. Sixteen dogs were divided into four groups of four and received costo-abdominal right-side full-thickness skin wounds. Thereafter, a bacterial suspension was administered to each wound bed. To assess wound infection, a complete bacterial load count was performed. Clinical wound healing assessments were done (healing rate (%), healing time (days), and comparisons between AgNPs and Amikacin were made to determine their antibacterial actions, as well as histological examination of the skin. *Tribulus terrestris* was used for the biosynthesis of AgNPs in this study. The AgNPs were characterized using UV-vis spectroscopy, SEM, XRD, EDS, AFM and zeta potential. AgNPs, AgNPs-Amikacin, Amikacin, or no treatment (control group) were topically administered to the wound bed. AgNPs cure wounds in 20 ± 1.64 days, while Amikacin takes 28 ± 1.75. In comparison to amikacin, AgNPs 18 ± 1.47 healed the fastest. AgNPs-Amikacin, AgNPs, and Amikacin can eradicate *P. aeruginosa*-infected wounds in 7, 9, and 14 days, according to bacterial counts. AgNPs reduce inflammation and increase collagen fiber deposition to treat skin lesions. AgNPs also reduced hemorrhagic areas and inflammatory cells in *P. aeruginosa*-infected wounds, aiding wound healing. All in all, these nanoparticles reduced bacterial wound infections and aided tissue healing. It has been found that AgNPs can be used as an efficacious antimicrobial and anti-inflammatory agent in wound healing instead of antibiotics.

## 1. Introduction

Wound healing is divided into four stages: hemostasis, inflammation, proliferation, and maturation. Several growth factors have been shown to be required for the beginning and promotion of wound healing [1].

**Data availability statement:** All relevant data are within the paper and its Supporting Information files.

**Funding:** The author(s) received no specific funding for this work.

Re-epithelialization, a critical process during the early stages of wound healing, happens not only by the migration and proliferation of keratinocytes in the epidermal layer of skin from the wound edge, but also by the differentiation of stem cells living in the bulge of hair follicles. Rapid re-epithelialization following wounding creates an ideal environment for healing, including a scaffold of cells and numerous growth factors that are essential in the wound-healing process [2].

The detrimental effect of some species of bacteria on wound healing has been intensively investigated, and these microorganisms are frequently identified as the cause of delayed wound healing. The most frequent bacterium responsible for these disorders is *Pseudomonas aeruginosa.* It is a Gram-negative, rod-shaped, non-spore-forming, encapsulated, and motile bacteria [3].

*Pseudomonas aeruginosa (P. aeruginosa)* is a saprophytic bacterium that is found in soil, water, plants, animals, and humans. Owing of the adaptability and intrinsic antibiotic resistance of *P. aeruginosa,* conventional antimicrobials, such as antibiotics, are generally ineffective, hence increasing mortality [4]. In addition, the ability of *P. aeruginosa* to create biofilms, which protects it from environmental stressors, inhibits phagocytosis, and imparts colonization and long-term persistence, hinders the treatment of these infections [5].

Nanotechnology is a fast-increasing discipline involving the building of novel materials between 1 and 100 nm with applications in a variety of fields, including research, agriculture, and anti-infective therapy. As a result of their extraordinary susceptibility to microbial infection, wounds pose extraordinarily challenging medical issues. Furthermore, wounds must heal fast and properly with little scarring. Nanoparticles provide a plethora of biological applications that assist the treatment of a number of wound types [6].

Silver nanoparticles (AgNPs) has been shown to be more effective than other nanoparticles in wound healing. It possesses antibacterial qualities, making it a more effective agent in wound treatment [7]. AgNPs are effective against *E. coli, S. aureus, Klebsiella,* and *Pseudomonas* bacteria. In low concentrations, they are safe to humans despite causing cell death by attacking the respiratory chain and cell division [8]. Amikacin is a bactericidal, poorly absorbed aminoglycoside antibiotic. Metal nanoparticles are hydrophobic, but Amikacin is hydrophilic. Thus, antibiotics linked to metal nanoparticles can be easily delivered to cells [9].

The current study investigated the effect of biosynthesized silver nanoparticles alone and in combination with Amikacin was evaluated on surgically induced full thickness skin wounds in dogs injected with *P. aeruginosa.*

## 2. Materials and methods

### 2.1. Ethical approval

This study was conducted in conformity with the rules of the Animal Ethics Committee of the Faculty of Veterinary Medicine at the University of Kufa (license number: 27735 in 13/12/2022).

### 2.2. Characterization of AgNPs

The utilization of bio-synthesized AgNPs generated by *Tribulus terrestris* was employed [10]. Characterization was performed using UV-visible spectroscopy, scanning electron microscope (SEM). X-ray diffraction (XRD), EDS, AFM, The Zeta potential. UV – Visible spectrophotometer analysis, the color shift of the reaction mixtures was tracked by evaluating the UV-visible spectra of the reaction mixture after diluting it with deionized distillation water on a regular basis [11].

SEM determined nanoparticle shape and size. X-ray diffraction (XRD) is used to analyze molecular and crystal morphologies, qualitative detection of chemicals, quantitative resolution of chemical species, crystallinity, isomorphous replacements, and particle sizes [12]. EDS was used to determine the presence of nanoparticles in a point analysis [11]. The dispersion and aggregation of nanomaterials, as well as their size and form, were investigated using AFM [13]. Zeta potential analyzer was used to measure the stability of produced nanoparticles from −160 mV to +160 mV and depict the data in graphs [14].

## 2.3. Preparation of bacteria

In the Al-Najaf Veterinary Hospital, *P. aeruginosa* was isolated from dogs with external otitis. The VITEK-2 compact automated system was utilized for bacterial identification and anti-biogram testing based on the MIC technique determination. In Muller-Hinton broth, the microorganisms were grown. The bacteria were diluted to a concentration of $1 \times 10^8$ CFU/ml in sterile phosphate-buffered saline after being centrifuged at 1,000 rpm for 15 minutes. After wound surgery, 100 microliters ($2.7 \times 10^6$ CFU) of the bacterial suspension should be added immediately to each wound bed [15].

## 2.4. Animals and experimental design

Sixteen healthy local breed dogs of both sexes (6–8 months old) were used for this experiment. The dogs were divided into four groups and each group comprising 4 dogs. They were separately housed in ($1 \times 1.25$) meter cages. The dogs are fed balanced diet and are supervised by a skilled staff. The experiment was conducted from December 2022 to February 2023 in the animal house of the faculty of Veterinary medicine at the University of Kufa, Iraq.

**2.4.1. Anesthesia and wounding.** The dogs were sedated intramuscularly with a Ketamine–Xylazine–saline solution comprising 15 mg/kg of Ketamine (Holland) and 5 mg/kg of Xylazine (Belgium). In brief, 2 cm in diameter full-thickness skin incisions were produced on the dog's costoabdominal area right side after dog's hair was removed and exposed skin was washed with 70% ethanol (Fig 1).

**2.4.2. Experimental Treatments.** As previously noted, the dogs were separated into four groups, and each group received the following medication four days after the onset of infection:

Group (A): Control positive/non-treated group.

Group (B): treated with AgNPs 16 mg/30 Gram Vaseline applied daily for 4 days on infected skin.

Group (C): treated with AgNPs (8 mg) mixed with Amikacin 2.5 gm/30 Gram Vaseline applied for 4 days on infected skin.

Group (D): treated with Amikacin 5 gm/30 Gram Vaseline applied for 4 days on infected skin.

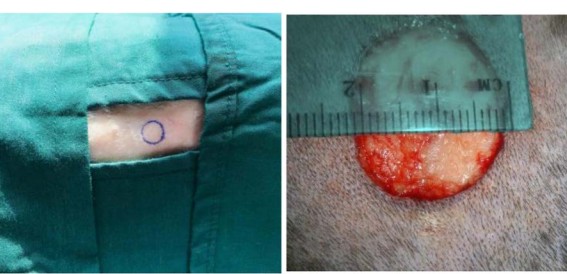

**Fig 1. A dog with incisions on its skin.**

### 2.5. Data collection

**2.5.1. Documentation of wound healing time.** At the moment of wound induction to re-epithelialization, the duration of wound healing was measured. All the observations were estimated randomly (7, 14, 21, and 28) days, until the scar disappeared [16].

**2.5.2. Determination of healing rate.** At 7-day intervals, the rate of healing for each treatment group was assessed. It was determined by dividing the highest average wound margin distance from the wound's center by the time required for wound closure. The following formula was used to determine the weekly healing rate:

Wound Closure (%) = [(Area of the wound on day 0 (mm) – area of the wound on indicated day) (mm)]/Area of the wound on day 0 (mm) × 100 [17]].

**2.5.3. Identification of WBC count.** WBC count were carried out by using the Vet. Scan HM5 hematology equipment (ABAXIS Company). Total Leukocyte Count (TLC) was performed before (0 time) and weekly after the induced infected wound.

**2.5.4. Microbiological evaluation.** Sterile swabs were collected from all infected wounds at 3-, 7- and 14-days post-wounding according to [18] for evaluation of bacterial load. The swab was rubbed gently over the wound center away from haired skin, and then put in a tube containing nutrient broth. Samples were incubated at 35Ċ overnight on Cetrimide agar medium. On the next day, bacterial colonies were counted, and bacterial load was calculated as the percentage of mean bacterial load of control wound.

**2.5.5. Histopathological Evaluation.** On days 7, 14, and 21 of the experiment, a histological test was done to discriminate between these groups by taking 5 mm of the healing regions. Overnight, the skin slices were fixed in a 10% formalin solution (pH 7.4). They were then processed through a succession of alcohol and xylene grades. Following that, the tissues were immersed in paraffin wax at 65 Ċ. Tissue blocks were cut into 5μm thick slices, stained with hematoxylin and eosin (H&E), and examined under a light microscope [19].

### 2.6. Statistical analysis

Data were investigated using SPSS version 26 and GraphPad Prism 8.0. The results were presented as mean ± standard errors (SE), and a P value < 0.05 was considered statistically significant. The LSD was used to find differences between the groups. The data were statistically analyzed using the one-way ANOVA.

## 3. Results

### 3.1. Characterization of AgNPs

UV-Visible spectrophotometry, which revealed that the nanoparticles have the largest absorption peak at wave lengths of (410 nm) as shown in Fig 2-A. XRD revealed that the size of AgNPs was 50.2 nm (Fig 2-B). Examining the existence of silver atoms using EDS (Fig 2-C). The Zeta potential of the particle size analyzer was 45 mV (Fig 2-D).

The SEM had a size range of 20 to 25 nm and was spherical and homogeneous (Fig 3-A). The AFM revealed the three-dimensional structure of 60.17 nm nanoparticles of silver (Fig 3-B).

### 3.2. Wound healing rate (%)

During the first to the fourth week of the research period, the healing rate of the wounded doss with *P. aeruginosa* increased gradually (P < 0.05) and varied significantly across the experimental treatments (Fig 4).

The AgNPs-Amikacin treatment group had the highest rate of healing (100.75 ± 1.41), followed by the AgNPs treatment group (97.50 ± 1.42) and the Amikacin treatment group

$(91.50 \pm 2.31)$. For the duration of the study, AgNPs-Amikacin treatment group performed the best in terms of healing rate (1st week to 4th week). These findings suggest that silver nanoparticles expedite the healing of wounds.

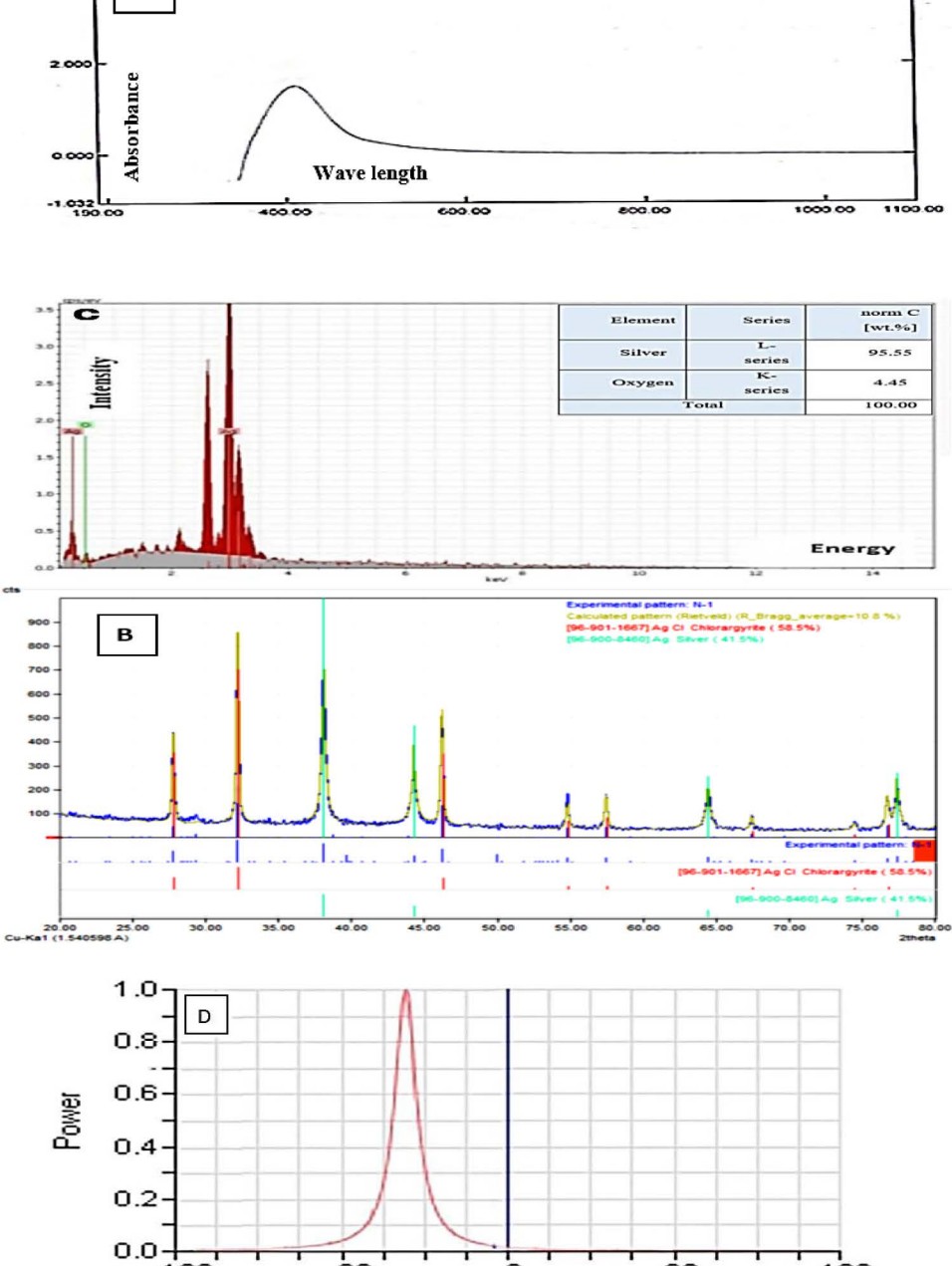

**Fig 2. (A) UV–visible absorption spectrum of AgNPs. The absorption spectra of AgNPs showed a significant wide peak at 410 nm. B: X-ray diffraction of silver nanoparticles produced from** *Tribulus terrestris* **extract.** C: Silver (95.55%) and oxygen (4.45%). D: Determination **of surface charge of biosynthesized silver nanoparticles by zeta potential.**

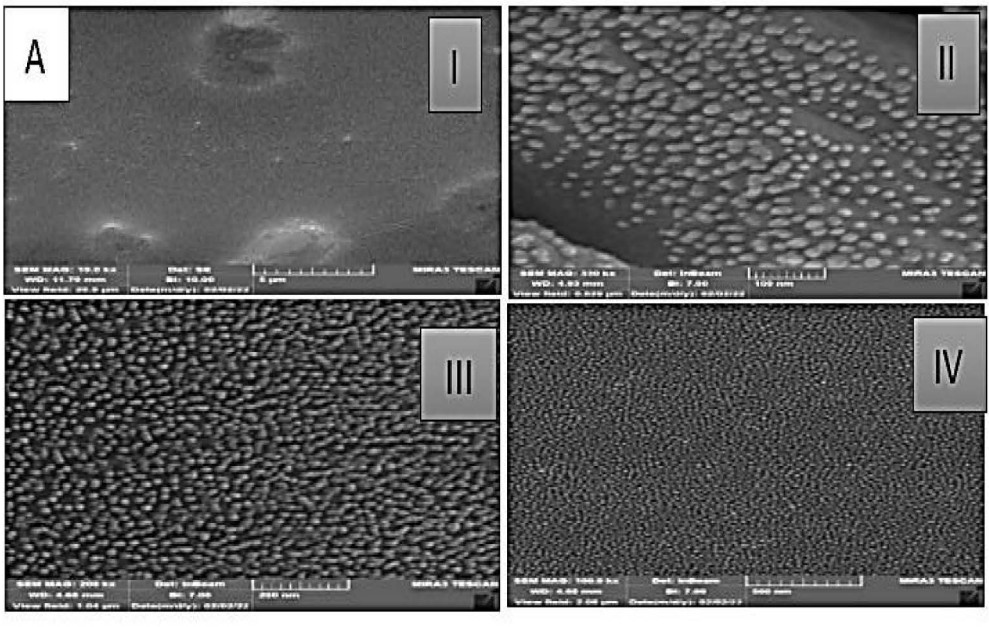

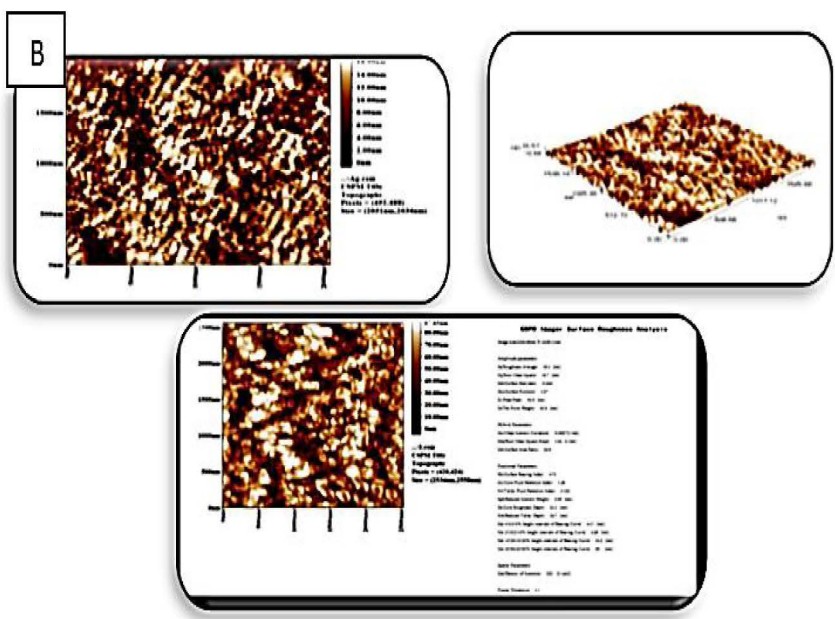

**Fig 3. A: Shows the SEM image of AgNPs image (I) at 5 μm, (II) under 100nm, (III) in 200nm and (IV) in 500 nm was obtained at 2000 x magnification indicates AgNPs are spherical in form with smooth surface and the particle size is about 20 -25 nm. B:** Atomic **Force Microscopic analysis of AgNPs produced by** *Tribulus terrestris* **(Average Diameter: 60.17 nm).**

### 3.3. Healing time

The study found a significant difference ($P < 0.05$) between experimental treatments. Group A (control), Group B (AgNPs), Group C (AgNPs + Amikacin), and Group D (Amikacin) had Healing times of $40 \pm 1.87$, $20 \pm 1.64$, $18 \pm 1.47$, and $28 \pm 1.75$, respectively. During the course

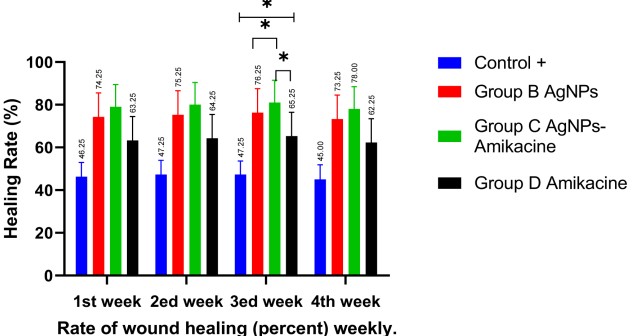

**Fig 4. The effects of AgNPs, AgNPs-Amikacin, and Amikacin on the rate of wound healing (percent) weekly.** *
**Values are significantly different under the probability level (P ≤ 0.05).**

of treatment, no adverse effects were seen in terms of the animals' body weight, overall health, or behavior. Regarding the total healing time, wounds treated with AgNPs and AgNPs-Amikacin healed faster than Amikacin and control wounds.

Fig 5 shows images of the groups of infected wounds treated with AgNPs for 0, 7, 14, and 21 days. Significantly, after 7 days of treatment, AgNPs successfully reduced the size and inflammation of the infected wound. Re-epithelialization is vital in wound healing as the skin performs a significant barrier function in protecting the body against pathogens.

After 3 and 7 days, the untreated, infected wounds of the control groups displayed significant inflammation (Fig 6).

### 3.4. WBC count

The WBC count of the various treatments studied was significantly affected by the experimental groups (P < 0.05), as shown in Fig 7.

The present study demonstrated that the WBC count was significantly increased in the non-treated/control group (at 0 and 28 days) compared to the treated group. Nevertheless, the treated group rebounded at 14-, 21-, and 28-day intervals, with the 7-day interval showing the greatest increase.

### 3.5. Microbiology

All of the treatment groups that received either AgNPs or Amikacin were shown to successfully inhibit the development of the bacteria *P. aeruginosa*. The results indicate that group C treatment could eradicate all bacterial cell-infected wounds in 7 days, whereas group B treatment could eradicate all bacterial cell-infected wounds in 9 days, and group D treatment could

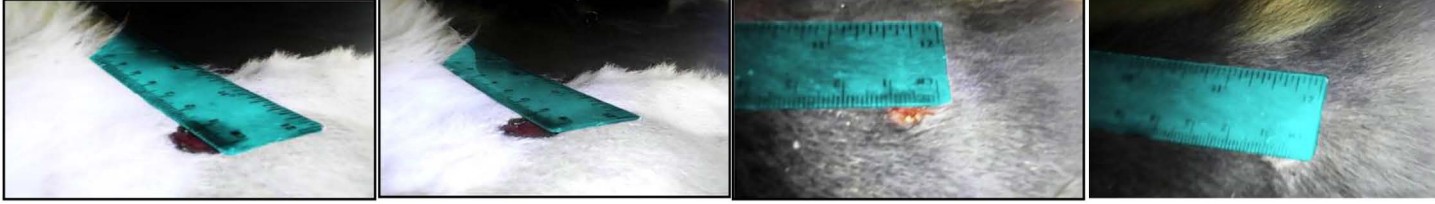

**Fig 5. (A) Digital images of wound healing taken on various days (0, 7, 14 and 21).**

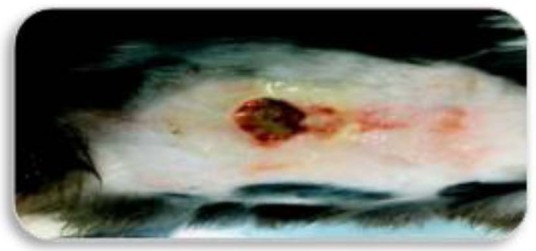

**Fig 6. Wounds infected in the control groups.**

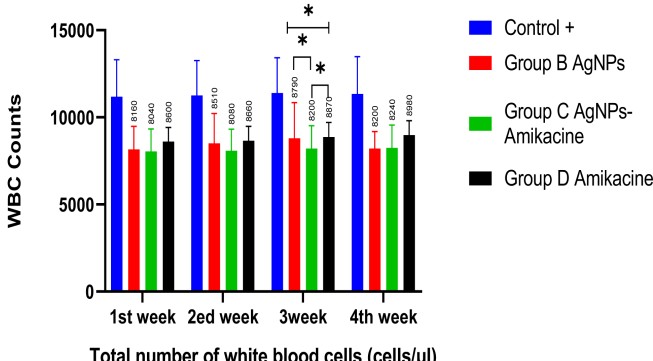

**Fig 7. The effects of AgNPs, AgNPs combined with Amikacin, and Amikacin on the Total number of white blood cells (cells/µl) in dogs at weekly intervals. * Values are significantly different under the probability level (P ≤ 0.05).**

eradicate all bacterial cell-infected wounds in 14 days (Table 1). This confirmed that AgNPs are a more effective antibacterial agent.

## 3.6. Histological examination

Fig 8 displays the findings of a histological study. Healed wounds from the AgNPs and AgNPs-Amikacin groups had nearly normal keratin and epidermal layers covering a dense layer of mature horizontally oriented granulation tissue and collagen matrix, with extensive formation of rete ridges, dermal glands, and hair follicles, and no scar.

**Table 1. Bacterial count prior to and following treatment (Means ± SE).**

| Treatment groups | 3 Days Prior TTT | 3 Days After TTT | 7 Days | 9 days | 12 days | 14 Days |
|---|---|---|---|---|---|---|
| Group A Control (+) | 109 ± 2.43Ba | 150 ± 3.43Ba | 220 ± 5.86 Aa | 252 ± 8.91Aa | 270 ± 3.75 Aa | 275 ± 6.47 A |
| Group B AgNPs | 115 ± 4.93 Aa | 75 ± 3.93 Bc | 12.5 ± 0.69Cc | 0 ± 0 | 0 ± 0 | 0 ± 0 |
| Group C AgNPs + Amikacin | 114 ± 1.61Aa | 25 ± 1.59Bb | 0 ± 0 | 0 ± 0 | 0 ± 0 | 0 ± 0 |
| Group D Amikacin | 120 ± 7.43Aa | 65 ± 1.88 Bd | 53 ± 2.94 Cb | 22 ± 1.58 Db | 10 ± 0.18Eb | 0 ± 0 |

Capital litters indicated that: Means within the same row bearing different litters are significantly different at (P < 0.05).

Small litters indicated that: Means within the same column bearing different litters are significantly different at (P < 0.05).

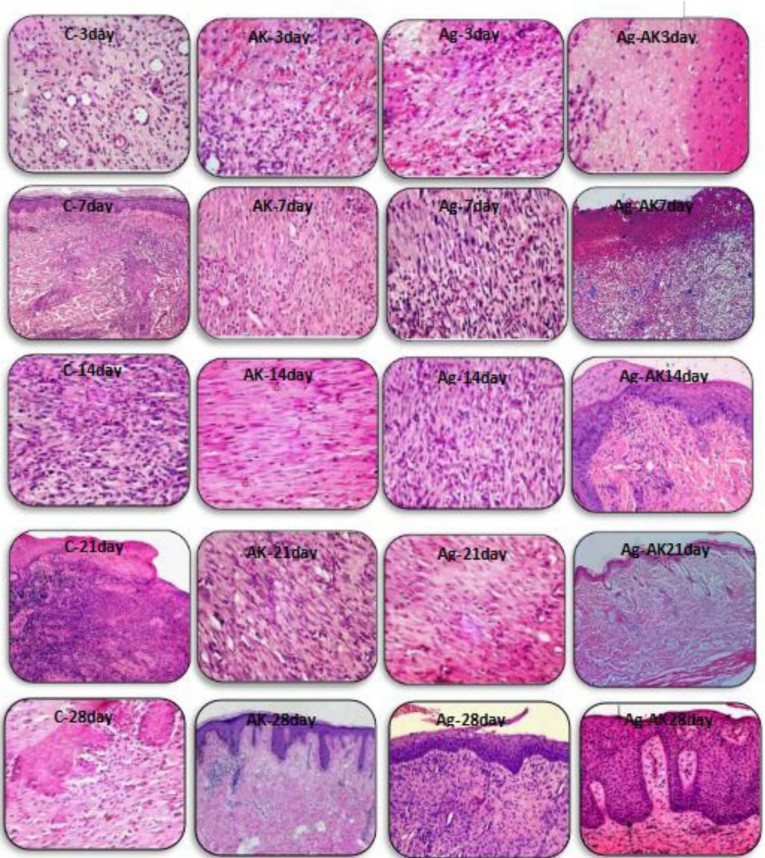

**Fig 8. Histological examination of dog wounds on the days 3, 7, 14, 21 and 28.** Control (C), Amikacin (AK), AgNPs (Ag), AgNPs-Amikacin (Ag-AK).

Histological images of Amikacin-treated wounds revealed thicker keratin and epidermal layers covering mature granulation tissue with mixed vertical and horizontal orientation, as well as modest to moderate production of rete ridges, dermal glands, and hair follicles. AgNPs clearly accelerated the pace of re-epithelialization, angiogenesis, fibroblast proliferation, and the creation of mature granulation tissues. Furthermore, AgNPs induced the early infiltration of polymorphic inflammatory cells, which decreased on day 7 to end the inflammatory phase quicker than control and Amikacin-treated wounds.

During the inflammatory phase (3 days post-wounding), control untreated wounds showed bleeding, edema, and infiltration of polymorphic inflammatory cells, whereas Amikacin-treated wounds showed significant inflammatory infiltrations and fibrin network development. Topical treatment of AgNPs and AgNPs-Amikacin treated wounds, on the other hand, resulted in the creation of a few newly created blood vessels as well as substantial inflammatory infiltration and proliferation of immature granulation tissues.

Control wounds showed limited proliferation of the covering epithelial cells, minor underlying neovascularization, and macrophage infiltration after 7 days. In Amikacin-treated wounds, however, practically complete re-epithelilization with substantial underlying neo-vascularization and decreased inflammatory response was seen. After topical treatment with AgNPs and AgNPs-Amikacin treated wounds, however, there was complete proliferation of the covering epithelium, extensive neovascularization, and the presence of a few fibroblast

and collagen fibers perpendicular to the newly formed blood vessels and parallel to the wound surface.

After 14 and 21 days, wounds from the control and Amikacin-treated groups showed little macrophage infiltration, full epidermal coverage, and replacement of the incisional region with mature vertically oriented granulation tissues. In AgNPs and AgNPs-Amikacin-treated wounds, however, full re-epithelialization and mixed orientation of mature granulation tissues were observed.

## 4. Discussion

### 4.1. WBC count

The Total number of white blood cells increased in the control group as a result of tissue destruction and microbial infiltration at the wound site. The inflammatory processes prolonged the inflammatory phase and raised the number of inflammatory cells for almost three weeks. Leukocyte levels in treated animals increased during the first week as a result of inflammation caused by tissue fragments. After fourteen days, their levels had returned to normal. This was due to AgNPs action as an anti-microorganism agent, as well as the fact that their contents enhanced construction and eliminated debris from the lesion region. These findings are comparable to those of [20], who concluded that wound dressings containing AgNPs can effectively prevent bacterial infection and stimulate tissue regeneration throughout the healing process.

### 4.2. Antibacterial activity of AgNPs

The powerful antibacterial capabilities of AgNPs have been extensively researched, with promising results for future antibacterial agents [21] and [22]. AgNPs have antimicrobial and antibiofilm properties against gram-negative bacteria, including Pseudomonas aeruginosa and multidrug-resistant strains. Their low concentrations can cause rapid cytotoxicity and death in microorganisms [23] and [24].

In comparison to Amikacin, silver nanoparticles significantly reduced bacterial load from wounds until 9 days after wounding and reduced wound healing time by an average of 8 days. These findings might be explained by two pieces of evidence.

First, bacterial resistance to Amikacin has already been described. The chemical change of aminoglycoside-modifying enzymes is the most prevalent mechanism of Amikacin resistance [25]. Secondly, in contrast, the mechanism of action of AgNPs in the cell of bacteria is that AgNPs adhere to cell membranes, aggregate, and modify the lipid bilayer, which ultimately results in an increase in the permeability of the lipid bilayer. Observations indicate that ions generated by silver oxidation permeate cells and interact with enzyme pathways, nucleic acids, and other cytoplasmic components, affecting intracellular signals and transduction pathways. In addition, the oxidation of silver and the release of Ag + might result in the formation of reactive oxygen species (ROS) and other radicals that are capable of changing many substances, including DNA, proteins, and lipids [26].

The antibacterial activity of AgNPs may be dependent on their concentration, duration of cell exposure, surface charge, shape, and notably size. Positively charged nanoparticles are believed to be the most efficient against gram-positive and gram-negative bacteria because they alter the membrane potential [27]. This study's findings are in accordance with those of [28], who discovered that AgNPs had antibacterial effect against bacteria *P. aeruginosa* [29]. postulated that the combination of AgNP with antimicrobials that work at the intracellular level may act synergistically because AgNP destabilizes the cell membrane and increases the internalization of the other medication, hence enabling access to its target. Others have

discovered that AgNPs have antibacterial and anti-inflammatory qualities when used to treat infected wounds [30].

### 4.3. Silver nanoparticles promote wound healing activity

It has previously been stated that silver enhances the healing process and speeds up all of the needed processes. Silver nanoparticles also speed up wound contraction, which is important in the healing process. Additionally, researchers revealed that AgNPs expedited re-epithelialization to improve wound healing *in vivo* and *in vitro* [31]. The current study demonstrates that AgNPs can accelerate the healing of skin lesions and reduce the visibility of scars.

AgNPs is most commonly used in medicine, and there is similarity with other researchers in the internal medicine and microbiology fields [32], who used AgNPs to promote wound healing and discovered AgNPs has a greater effect on wound healing than silver sulfadiazine. The wound healing response may be broken down into a series of distinct stages: the inflammatory stage, the proliferative stage, and the maturation stage. The immune system is most active during the phase of inflammation. Generally, the phases of wound healing involve inflammation at the site of damage, angiogenesis and the growth of granulation tissue, repair of the connective tissue and epithelium, and finally conversion to a healed wound [33].

We demonstrate in this study the in vivo capabilities of AgNPs that appear to expedite the healing of wounds in a dog model of skin wound that were inoculated with *P. aeruginosa.* Apparently, AgNPs are responsible for reducing the time required for hyperactive cells (myofibroblasts) to create contractile force in a wound and for reversing inflammatory processes more effectively than antibiotics and this is in accordance with the findings of a study [34].

### 4.4. Histological examination

Our pathology findings following *P. aeruginosa* infected wound treatment with AgNPs indicate enhanced histological changes in wound healing tissue, such as granulation early formation and maturation. After 7 days of treatment with AgNPs, the tissue structure in the repaired area was restored, revealing stratified epidermis with granular and cornified layers, indicating the efficiency of AgNPs in wound healing. Our findings are consistent with those of others [35]; stated that AgNPs might accelerate granulation tissue production and maturation, as well as the progression of the primary collagen scar and rudimentary cutaneous appendages.

## 5. Conclusions

Nano silver could provide an alternate therapy for wounds infected with *P. aeruginosa,* demonstrating AgNPs' ability to promote healing through antibacterial and anti-inflammatory capabilities. The study discovered that when Amikacin and silver nanoparticles are used together to cure a wound, they have a synergistic impact. AgNPs mixed with Amikacin resulted in quicker wound healing with fewer minimum necessary days and no evident adverse effects, indicating that wound healing is possible without the use of antibiotics like Amikacin.

On a histological level, the AgNP-treated group healed faster, with complete epidermal re-epithelialization, substantial fibroblastic activity, collagen-rich dermal granulation tissue formation, and low inflammatory cell infiltration.

### Acknowledgments

The authors appreciate the moral assistance of the Veterinary Medicines/University of Kufa, Iraq.

## Author contributions

**Conceptualization:** Ali Hussein Aldujaily.

**Data curation:** Ali Hussein Aldujaily.

**Formal analysis:** Ali Hussein Aldujaily.

**Funding acquisition:** Ali Hussein Aldujaily.

**Investigation:** Kifah Fadhil Hassoon.

**Methodology:** Kifah Fadhil Hassoon.

**Project administration:** Kifah Fadhil Hassoon.

**Software:** Murtadha Abbas.

**Supervision:** Abdulameer Abid Hatem.

**Visualization:** Murtadha Abbas, Abdulameer Abid Hatem.

**Writing – original draft:** Murtadha Abbas.

**Writing – review & editing:** Murtadha Abbas.

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
