## [Decision Letter · Decision Letter 0]

15 May 2024

PONE-D-24-01782The Effect of Biosynthesized Silver Nanoparticles on Pseudomonas aeruginosa-Infected Dogs Wounds.PLOS ONE

Dear Dr. Abbas,

Thank you for submitting your manuscript to PLOS ONE. After careful consideration, we feel that it has merit but does not fully meet PLOS ONE’s publication criteria as it currently stands. Therefore, we invite you to submit a revised version of the manuscript that addresses the points raised during the review process.

We look forward to receiving your revised manuscript.

Kind regards,

Abeer El Wakil, PhD

Academic Editor

PLOS ONE

Journal Requirements:

2. Please amend the manuscript submission data (via Edit Submission) to include author Ali Hussein Aldujaily.

3. We note that Figure(s) 1, 3, 6, 8, 9, 10, 11 and 12 in your submission contain copyrighted images. All PLOS content is published under the Creative Commons Attribution License (CC BY 4.0), which means that the manuscript, images, and Supporting Information files will be freely available online, and any third party is permitted to access, download, copy, distribute, and use these materials in any way, even commercially, with proper attribution. For more information, see our copyright guidelines: http://journals.plos.org/plosone/s/licenses-and-copyright.

a. You may seek permission from the original copyright holder of Figure(s) 1, 3, 6, 8, 9, 10, 11 and 12 to publish the content specifically under the CC BY 4.0 license. 

Reviewers' comments:

Reviewer's Responses to Questions

**Comments to the Author**

1. Is the manuscript technically sound, and do the data support the conclusions?

Reviewer #1: Partly

Reviewer #2: Yes

2. Has the statistical analysis been performed appropriately and rigorously? 

Reviewer #1: Yes

Reviewer #2: Yes

3. Have the authors made all data underlying the findings in their manuscript fully available?

Reviewer #1: Yes

Reviewer #2: Yes

4. Is the manuscript presented in an intelligible fashion and written in standard English?

Reviewer #1: No

Reviewer #2: Yes

5. Review Comments to the Author

Reviewer #1: This study evaluates the use of biosynthetic AgNPs in combination or not with amikacin in a Pseudomonas aeruginosa-infected canine wound model in vivo. The article is not very innovative, but the in vivo model is not very common. Some modifications are needed before considering it in more detail.

Important notes

The English language needs to be checked by a professional translator. This is urgently needed.

The Materials and methods section needs to be clarified. What happened to the dogs after the in vivo model? Could you please specify the breed of the dogs to ensure reproducibility? What happened to the infection control dogs after the in vivo model? Did the dogs wear an Elizabethan collar after treatment? How were the animals housed? Describe the facilities where the animals were housed?

Tables 1 and 3 can be presented graphically. This way it would be more visual and intuitive. The data from Table 1 should be analysed using a linear correlation over time and the slope can be compared pairwise. The data from Table 3 should be analysed using a Spearman correlation over time.

Table 2 can be summarised in a few lines in the main text and removed.

I miss a collage with pictures of at least one wound from each treatment group over time. Could the author add this?

Figures 10, 11 and 12 can be combined into a single collage. Please make sure that you provide clear and nice images with good quality and one scale per image for each condition.

In the Discussion section, lines 385-416 are a repeat of the Introduction section. Please delete these lines and add a first paragraph describing the main findings of your study.

The Discussion section lacks a comparison between the proposed therapeutic strategy and the other therapeutic AgNP- or NP-only based strategies available in the literature. Please add this information.

Reviewer #2: The manuscript by Aldujaily and coworkers is an interesting proof of concept of the ability of silver nanoparticles to promote curing of wounds infected by P. aeruginosa in dogs.

The study is well designed and the conclusions obtained valid; nevertheless I have some concerns:

Major:

1) Only one strain of P. aeruginosa was used (an isolate from dog otitis, which is a good selection) and variability of strains may lead to different outcomes.

2) The antibiotic susceptibility profile of the isolate should be shown.

3) The antibacterial effect of the nanoparticles in vitro should be shown, if possible against several strains.

4) Despite of the existence of several papers showing the effectivity of silver nanoparticles against clinical isolates of P. aeruginosa, those were not mentioned in the introduction of discussion:

Salomoni R, Léo P, Montemor AF, Rinaldi BG, Rodrigues M. Antibacterial effect of silver nanoparticles in Pseudomonas aeruginosa. Nanotechnol Sci Appl. 2017 Jun 29;10:115-121. doi: 10.2147/NSA.S133415. PMID: 28721025; PMCID: PMC5499936.

Al-Momani H, Almasri M, Al Balawi D, Hamed S, Albiss BA, Aldabaibeh N, Ibrahim L, Albalawi H, Al Haj Mahmoud S, Khasawneh AI, Kilani M, Aldhafeeri M, Bani-Hani M, Wilcox M, Pearson J, Ward C. The efficacy of biosynthesized silver nanoparticles against Pseudomonas aeruginosa isolates from cystic fibrosis patients. Sci Rep. 2023 Jun 1;13(1):8876. doi: 10.1038/s41598-023-35919-6. PMID: 37264060; PMCID: PMC10235065.

The potential activity of biosynthesized silver nanoparticles of Pseudomonas aeruginosa as an antibacterial agent against multidrug-resistant isolates from intensive care unit and anticancer agent

• A. B. Abeer Mohammed,

• Mona Mohamed Abd Elhamid,

• Magdy Kamal Mohammed Khalil,

• Abdallah Soubhy Ali &

• Rateb Nabil Abbas

Environmental Sciences Europe volume 34,

Campo-Beleño C, Villamizar-Gallardo RA, López-Jácome LE, González EE, Muñoz-Carranza S, Franco B, Morales-Espinosa R, Coria-Jimenez R, Franco-Cendejas R, Hernández-Durán M, Lara-Martínez R, Jiménez-García LF, Fernández-Presas AM, García-Contreras R. Biologically synthesized silver nanoparticles as potent antibacterial effective against multidrug-resistant Pseudomonas aeruginosa. Lett Appl Microbiol. 2022 Sep;75(3):680-688. doi: 10.1111/lam.13759. Epub 2022 Jun 22. PMID: 35687297; PMCID: PMC9543579.

Please cite them.

Minor:

1) L 34-37 “AgNPs cure wounds in 20 days, while Amikacin takes 28. In comparison to amikacin, AgNPs healed the fastest. AgNPs-Amikacin, AgNPs, and Amikacin can eradicate P. aeruginosa infected wounds in 7, 9, and 14 days, according to bacterial counts.”

Add standard deviation to these values.

2) Pseudomonas aeruginosa is mentioned in line 59 for the1rst time (use P. aeruginosa from this line).

3) Line 152 (7, 14, 21, and 28), days?

4) L 362 “ontrol”

5) Put latin expressions in italics : in situ, in vivo, in vitro, etc.

6) Capitalize “Gram”

7) All scientific and gene names in the references should be in italics

6. PLOS authors have the option to publish the peer review history of their article (what does this mean? ). If published, this will include your full peer review and any attached files.

**Do you want your identity to be public for this peer review?** For information about this choice, including consent withdrawal, please see our Privacy Policy .

Reviewer #1: No

Reviewer #2: No

---

## [Author Response · Author response to Decision Letter 0]

23 Jun 2024

Revise the current sentence. Reviewer 1

Line No. Question Answer

What happened to the dogs after the in vivo model?

The dogs were donated to certain breeders in rural areas to act as guards.

Could you please specify the breed of the dogs to ensure reproducibility?

All dogs involved in the trial were local breeds.

What happened to the infection control dogs after the in vivo model?

The dogs received special care in which the wounds were cleaned with alcohol and local and systemic antibiotics were administered.

Did the dogs wear an Elizabethan collar after treatment?

yes

How were the animals housed?

They were housed in (1×1.25) meters cages individually.

Describe the facilities where the animals were housed?

The animals were housed in the animal house found at the College of Veterinary Medicine.

Table 2 can be summarized in a few lines in the main text and removed.

The study found a significant difference (P<0.05) between experimental treatments. Group A (control), Group B (AgNPs), Group C (AgNPs+Amikacin), and Group D (Amikacin) had Healing times of 40 ±1.87, 20 ±1.64, 18 ±1.47, and 28 ±1.75, respectively.

In the Discussion section, lines 385-416 are a repeat of the Introduction section. Please delete these lines and add a first paragraph describing the main findings of your study.

Deleted

Reviewer 2

Line No. Question Answer

L 34-37 “AgNPs cure wounds in 20 ± 1.64 days, while Amikacin takes 28 ± 1.75. In comparison to amikacin, AgNPs 18 ± 1.47 healed the fastest. AgNPs-Amikacin, AgNPs, and Amikacin can eradicate P. aeruginosa infected wounds in 7, 9, and 14 days, according to bacterial counts.”

line 59 P. aeruginosa

Line 152 All the observations were estimated randomly (7, 14, 21, and 28), days

until the scar disappeared.

L 362 Control wounds

in situ, in vivo, in vitro, etc

Gram

---

## [Decision Letter · Decision Letter 1]

15 Jul 2024

PONE-D-24-01782R1The Effect of Biosynthesized Silver Nanoparticles on Pseudomonas aeruginosa-Infected Dogs Wounds.PLOS ONE

Dear Dr. Abbas,

Thank you for submitting your manuscript to PLOS ONE. After careful consideration, we feel that it has merit but does not fully meet PLOS ONE’s publication criteria as it currently stands. Therefore, we invite you to submit a revised version of the manuscript that addresses the points raised during the review process.

We look forward to receiving your revised manuscript.

Kind regards,

Abeer El Wakil, PhD

Academic Editor

PLOS ONE

Journal Requirements:

Reviewers' comments:

Reviewer's Responses to Questions

**Comments to the Author**

1. If the authors have adequately addressed your comments raised in a previous round of review and you feel that this manuscript is now acceptable for publication, you may indicate that here to bypass the “Comments to the Author” section, enter your conflict of interest statement in the “Confidential to Editor” section, and submit your "Accept" recommendation.

Reviewer #1: All comments have been addressed

Reviewer #2: (No Response)

2. Is the manuscript technically sound, and do the data support the conclusions?

Reviewer #1: Yes

Reviewer #2: Yes

3. Has the statistical analysis been performed appropriately and rigorously? 

Reviewer #1: Yes

Reviewer #2: Yes

4. Have the authors made all data underlying the findings in their manuscript fully available?

Reviewer #1: Yes

Reviewer #2: Yes

5. Is the manuscript presented in an intelligible fashion and written in standard English?

Reviewer #1: Yes

Reviewer #2: Yes

6. Review Comments to the Author

**Reviewer #1: ** Please, include all the answers given to me in the manuscript. They are important for the reproducibility of this study.

**Reviewer #2:**  I do not understand why you didnt replied most of my previous comments.

Please answer ALL THIS POINTS one by one:

Major:

1) Only one strain of P. aeruginosa was used (an isolate from dog otitis, which is a good selection) and variability of strains may lead to different outcomes.

2) The antibiotic susceptibility profile of the isolate should be shown.

3) The antibacterial effect of the nanoparticles in vitro should be shown, if possible against several strains.

4) Despite of the existence of several papers showing the effectivity of silver nanoparticles against clinical isolates of P. aeruginosa, those were not mentioned in the introduction of discussion:

Salomoni R, Léo P, Montemor AF, Rinaldi BG, Rodrigues M. Antibacterial effect of silver nanoparticles in Pseudomonas aeruginosa. Nanotechnol Sci Appl. 2017 Jun 29;10:115-121. doi: 10.2147/NSA.S133415. PMID: 28721025; PMCID: PMC5499936.

Al-Momani H, Almasri M, Al Balawi D, Hamed S, Albiss BA, Aldabaibeh N, Ibrahim L, Albalawi H, Al Haj Mahmoud S, Khasawneh AI, Kilani M, Aldhafeeri M, Bani-Hani M, Wilcox M, Pearson J, Ward C. The efficacy of biosynthesized silver nanoparticles against Pseudomonas aeruginosa isolates from cystic fibrosis patients. Sci Rep. 2023 Jun 1;13(1):8876. doi: 10.1038/s41598-023-35919-6. PMID: 37264060; PMCID: PMC10235065.

The potential activity of biosynthesized silver nanoparticles of Pseudomonas aeruginosa as an antibacterial agent against multidrug-resistant isolates from intensive care unit and anticancer agent

• A. B. Abeer Mohammed,

• Mona Mohamed Abd Elhamid,

• Magdy Kamal Mohammed Khalil,

• Abdallah Soubhy Ali &

• Rateb Nabil Abbas

Environmental Sciences Europe volume 34,

Campo-Beleño C, Villamizar-Gallardo RA, López-Jácome LE, González EE, Muñoz-Carranza S, Franco B, Morales-Espinosa R, Coria-Jimenez R, Franco-Cendejas R, Hernández-Durán M, Lara-Martínez R, Jiménez-García LF, Fernández-Presas AM, García-Contreras R. Biologically synthesized silver nanoparticles as potent antibacterial effective against multidrug-resistant Pseudomonas aeruginosa. Lett Appl Microbiol. 2022 Sep;75(3):680-688. doi: 10.1111/lam.13759. Epub 2022 Jun 22. PMID: 35687297; PMCID: PMC9543579.

Please cite them.

Minor:

1) L 34-37 “AgNPs cure wounds in 20 days, while Amikacin takes 28. In comparison to amikacin, AgNPs healed the fastest. AgNPs-Amikacin, AgNPs, and Amikacin can eradicate P. aeruginosa infected wounds in 7, 9, and 14 days, according to bacterial counts.”

Add standard deviation to these values.

2) Pseudomonas aeruginosa is mentioned in line 59 for the1rst time (use P. aeruginosa from this line).

3) Line 152 (7, 14, 21, and 28), days?

4) L 362 “ontrol”

5) Put latin expressions in italics : in situ, in vivo, in vitro, etc.

6) Capitalize “Gram”

7) All scientific and gene names in the references should be in italics

7. PLOS authors have the option to publish the peer review history of their article (what does this mean? ). If published, this will include your full peer review and any attached files.

**Do you want your identity to be public for this peer review?** For information about this choice, including consent withdrawal, please see our Privacy Policy .

Reviewer #1: No

Reviewer #2: No

---

## [Author Response · Author response to Decision Letter 1]

22 Jul 2024

Major:

1) Only one strain of P. aeruginosa was used (an isolate from dog otitis, which is a good selection) and variability of strains may lead to different outcomes.

Answer: I do not believe that employing more than one strain of P. aeruginosa will impact the results because pathogenic germs have not yet developed resistance to silver nanoparticles, and silver nanoparticles can eradicate pathogenic germs that are resistant and non-resistant to antibiotics.

2) The antibiotic susceptibility profile of the isolate should be shown.

Answer:

3) The antibacterial effect of the nanoparticles in vitro should be shown, if possible, against several strains.

Answer: Antibacterial activity of silver nanoparticles. Cultured on MHA at 37°C for 24 hrs with concentration of AgNPs (10, 20,40 and 80 μg/ml).

4) Despite of the existence of several papers showing the effectivity of silver nanoparticles against clinical isolates of P. aeruginosa, those were not mentioned in the introduction of discussion:

Answer: added in Line (390 - 394).

Minor

Reviewer 2

Line No.

Question

Answer

L 34-37

“AgNPs cure wounds in 20 ± 1.64 days, while Amikacin takes 28 ± 1.75. In comparison to amikacin, AgNPs 18 ± 1.47 healed the fastest. AgNPs-Amikacin, AgNPs, and Amikacin can eradicate P. aeruginosa infected wounds in 7, 9, and 14 days, according to bacterial counts.”

line 59

P. aeruginosa

Line 152

All the observations were estimated randomly (7, 14, 21, and 28), days

until the scar disappeared.

L 362

Control wounds

in situ, in vivo, in vitro, etc

Gram

---

## [Decision Letter · Decision Letter 2]

7 Aug 2024

PONE-D-24-01782R2The Effect of Biosynthesized Silver Nanoparticles on Pseudomonas erogenous-Infected Dogs Wounds.PLOS ONE

Dear Dr. Aldujaily,

Thank you for submitting your manuscript to PLOS ONE. After careful consideration, we feel that it has merit but does not fully meet PLOS ONE’s publication criteria as it currently stands. Therefore, we invite you to submit a revised version of the manuscript that addresses the points raised during the review process.

We look forward to receiving your revised manuscript.

Kind regards,

Abeer El Wakil, PhD

Academic Editor

PLOS ONE

Journal Requirements:

Additional Editor Comments :

The authors addressed successfully all reviewers' concerns. In my opinion, the work provides an advance towards the current knowledge about nanotechnology and bacterial wound healing. However, the quality of the presentation of the figures does not meet the standards set forth by the journal Plos One and I suggest publishing the paper in the journal with minor changes.

Here are my comments to the authors:

- The figures are appropriate, and understandable. However, they represent the major weak point of this manuscript. They are not presented in an insightful way. Some figures could be grouped, for example, Fig.2 + Fig.4 + Fig.5 (the Table should be minimized and adjusted within the figure) + Fig.7 could be grouped together. Fig.3+ Fig.6 could be grouped together. Also, the labels on Figure 6 are not clear and there is internal redundancy within the Figure.

- The spaces between the parts of Figure 12 should be minimized to be clear for readers.

- The citation of the figures within the text should be adjusted following figures' modification.

Good continuation.

Abeer El Wakil, PhD.

Reviewers' comments:

Reviewer's Responses to Questions

**Comments to the Author**

1. If the authors have adequately addressed your comments raised in a previous round of review and you feel that this manuscript is now acceptable for publication, you may indicate that here to bypass the “Comments to the Author” section, enter your conflict of interest statement in the “Confidential to Editor” section, and submit your "Accept" recommendation.

Reviewer #1: All comments have been addressed

Reviewer #2: All comments have been addressed

2. Is the manuscript technically sound, and do the data support the conclusions?

Reviewer #1: Yes

Reviewer #2: Yes

3. Has the statistical analysis been performed appropriately and rigorously? 

Reviewer #1: Yes

Reviewer #2: Yes

4. Have the authors made all data underlying the findings in their manuscript fully available?

Reviewer #1: Yes

Reviewer #2: Yes

5. Is the manuscript presented in an intelligible fashion and written in standard English?

Reviewer #1: Yes

Reviewer #2: Yes

6. Review Comments to the Author

Reviewer #1: (No Response)

Reviewer #2: Thanks for addressing my comments. I think that your point of all bacteria being susceptible to silver nanoparticles is dabatable and should be addressed experimentally. However I think even if only one strain was used in your work it is valuable enough to be accepted.

Minor: capitalize "Gram" it is a surname. put all scientific names in italics.

7. PLOS authors have the option to publish the peer review history of their article (what does this mean? ). If published, this will include your full peer review and any attached files.

**Do you want your identity to be public for this peer review?** For information about this choice, including consent withdrawal, please see our Privacy Policy .

Reviewer #1: No

Reviewer #2: No

---

## [Author Response · Author response to Decision Letter 2]

11 Sep 2024

Major:

1) Only one strain of P. aeruginosa was used (an isolate from dog otitis, which is a good selection) and variability of strains may lead to different outcomes.

Answer: I do not believe that employing more than one strain of P. aeruginosa will impact the results because pathogenic germs have not yet developed resistance to silver nanoparticles, and silver nanoparticles can eradicate pathogenic germs that are resistant and non-resistant to antibiotics.

2) The antibiotic susceptibility profile of the isolate should be shown.

Answer:

3) The antibacterial effect of the nanoparticles in vitro should be shown, if possible, against several strains.

Answer: Antibacterial activity of silver nanoparticles. Cultured on MHA at 37°C for 24 hrs with concentration of AgNPs (10, 20,40 and 80 μg/ml).

4) Despite of the existence of several papers showing the effectivity of silver nanoparticles against clinical isolates of P. aeruginosa, those were not mentioned in the introduction of discussion:

Answer: added in Line (390 - 394).

Minor

Reviewer 2

Line No. Question Answer

L 34-37 “AgNPs cure wounds in 20 ± 1.64 days, while Amikacin takes 28 ± 1.75. In comparison to amikacin, AgNPs 18 ± 1.47 healed the fastest. AgNPs-Amikacin, AgNPs, and Amikacin can eradicate P. aeruginosa infected wounds in 7, 9, and 14 days, according to bacterial counts.”

line 59 P. aeruginosa

Line 152 All the observations were estimated randomly (7, 14, 21, and 28), days

until the scar disappeared.

L 362 Control wounds

in situ, in vivo, in vitro, etc

Gram

---

## [Editor Report · Decision Letter 3]

13 Sep 2024

PONE-D-24-01782R3The Effect of Biosynthesized Silver Nanoparticles on Pseudomonas erogenous-Infected Dogs Wounds.PLOS ONE

Dear Dr.  Aldujaily,

Thank you for submitting your manuscript to PLOS ONE. After careful consideration, we feel that it has merit but does not fully meet PLOS ONE’s publication criteria as it currently stands. Therefore, we invite you to submit a revised version of the manuscript that addresses the points raised during the review process.

We look forward to receiving your revised manuscript.

Kind regards,

Abeer El Wakil, PhD

Academic Editor

PLOS ONE

Journal Requirements:

Additional Editor Comments:

Editor's comments:

The authors addressed successfully all reviewers' concerns. In my opinion, the work provides an advance towards the current knowledge about nanotechnology and bacterial wound healing. However, the quality of the presentation of the figures does not meet the standards set forth by the journal Plos One and I suggest publishing the paper in the journal with minor changes.

Here are my comments to the authors:

-      The figures are appropriate, and understandable. However, they represent the major weak point of this manuscript. They are not presented in an insightful way. Some figures could be grouped, for example, Fig.2 + Fig.4 + Fig.5 (the Table should be minimized and adjusted within the figure) + Fig.7 could be grouped together. Fig.3+ Fig.6 could be grouped together. Also, the labels on Figure 6 are not clear and there is internal redundancy within the Figure.

- The spaces between the parts of Figure 12 should be minimized to be clear for readers.

- The citation of the figures within the text should be adjusted following figures' modification.

---

## [Author Response · Author response to Decision Letter 3]

18 Sep 2024

Major:

1) Only one strain of P. aeruginosa was used (an isolate from dog otitis, which is a good selection) and variability of strains may lead to different outcomes.

Answer: I do not believe that employing more than one strain of P. aeruginosa will impact the results because pathogenic germs have not yet developed resistance to silver nanoparticles, and silver nanoparticles can eradicate pathogenic germs that are resistant and non-resistant to antibiotics.

2) The antibiotic susceptibility profile of the isolate should be shown.

Answer:

3) The antibacterial effect of the nanoparticles in vitro should be shown, if possible, against several strains.

Answer: Antibacterial activity of silver nanoparticles. Cultured on MHA at 37°C for 24 hrs with concentration of AgNPs (10, 20,40 and 80 μg/ml).

4) Despite of the existence of several papers showing the effectivity of silver nanoparticles against clinical isolates of P. aeruginosa, those were not mentioned in the introduction of discussion:

Answer: added in Line (390 - 394).

Minor

Reviewer 2

Line No. Question Answer

L 34-37 “AgNPs cure wounds in 20 ± 1.64 days, while Amikacin takes 28 ± 1.75. In comparison to amikacin, AgNPs 18 ± 1.47 healed the fastest. AgNPs-Amikacin, AgNPs, and Amikacin can eradicate P. aeruginosa infected wounds in 7, 9, and 14 days, according to bacterial counts.”

line 59 P. aeruginosa

Line 152 All the observations were estimated randomly (7, 14, 21, and 28), days

until the scar disappeared.

L 362 Control wounds

in situ, in vivo, in vitro, etc

Gram

---

## [Editor Report · Decision Letter 4]

14 Oct 2024

PONE-D-24-01782R4The Effect of Biosynthesized Silver Nanoparticles on Pseudomonas erogenous-Infected Dogs Wounds.PLOS ONE

Dear Dr. Aldujaily,

Thank you for submitting your manuscript to PLOS ONE. After careful consideration, we feel that it has merit but does not fully meet PLOS ONE’s publication criteria as it currently stands. Therefore, we invite you to submit a revised version of the manuscript that addresses the points raised during the review process.

Here are my comments to the authors:

-      The figures are appropriate, and understandable. However, they represent the major weak point of this manuscript. They are not presented in an insightful way. Some figures could be grouped, for example, Fig.2 + Fig.4 + Fig.5 (the Table should be minimized and adjusted within the figure) + Fig.7 could be grouped together. Fig.3+ Fig.6 could be grouped together. Also, the labels on Figure 6 are not clear and there is internal redundancy within the Figure.

- The spaces between the parts of Figure 12 should be minimized to be clear for readers.

- The citation of the figures within the text should be adjusted following figures' modification.

We look forward to receiving your revised manuscript.

Kind regards,

Abeer El Wakil, PhD

Academic Editor

PLOS ONE
---

## [Author Response · Author response to Decision Letter 4]

17 Oct 2024

Major:

1) Only one strain of P. aeruginosa was used (an isolate from dog otitis, which is a good selection) and variability of strains may lead to different outcomes.

Answer: I do not believe that employing more than one strain of P. aeruginosa will impact the results because pathogenic germs have not yet developed resistance to silver nanoparticles, and silver nanoparticles can eradicate pathogenic germs that are resistant and non-resistant to antibiotics.

2) The antibiotic susceptibility profile of the isolate should be shown.

Answer:

3) The antibacterial effect of the nanoparticles in vitro should be shown, if possible, against several strains.

Answer: Antibacterial activity of silver nanoparticles. Cultured on MHA at 37°C for 24 hrs with concentration of AgNPs (10, 20,40 and 80 μg/ml).

4) Despite of the existence of several papers showing the effectivity of silver nanoparticles against clinical isolates of P. aeruginosa, those were not mentioned in the introduction of discussion:

Answer: added in Line (390 - 394).

Minor

Reviewer 2

Line No. Question Answer

L 34-37 “AgNPs cure wounds in 20 ± 1.64 days, while Amikacin takes 28 ± 1.75. In comparison to amikacin, AgNPs 18 ± 1.47 healed the fastest. AgNPs-Amikacin, AgNPs, and Amikacin can eradicate P. aeruginosa infected wounds in 7, 9, and 14 days, according to bacterial counts.”

line 59 P. aeruginosa

Line 152 All the observations were estimated randomly (7, 14, 21, and 28), days

until the scar disappeared.

L 362 Control wounds

in situ, in vivo, in vitro, etc

Gram

---

## [Editor Report · Decision Letter 5]

21 Oct 2024

The Effect of Biosynthesized Silver Nanoparticles on Pseudomonas erogenous-Infected Dogs Wounds.

PONE-D-24-01782R5

Dear Dr. Aldujaily,

We’re pleased to inform you that your manuscript has been judged scientifically suitable for publication and will be formally accepted for publication once it meets all outstanding technical requirements.

Kind regards,

Abeer El Wakil, PhD

Academic Editor

PLOS ONE
---

## [Editor Report · Acceptance letter]

PONE-D-24-01782R5

PLOS ONE

Dear Dr. Aldujaily,

I'm pleased to inform you that your manuscript has been deemed suitable for publication in PLOS ONE. Congratulations! Your manuscript is now being handed over to our production team.

Kind regards,

on behalf of

Professor Abeer El Wakil

Academic Editor

PLOS ONE